# Neuronal transcriptome analyses reveal novel neuropeptide modulators of excitation and inhibition imbalance in *C. elegans*

**Katherine A. McCulloch, Kingston Zhou, Yishi Jin** *

Division of Biological Sciences, Section of Neurobiology, University of California San Diego, La Jolla, California, United States of America

* yijin@ucsd.edu

## Abstract

Neuropeptides are secreted molecules that have conserved roles modulating many processes, including mood, reproduction, and feeding. Dysregulation of neuropeptide signaling is also implicated in neurological disorders such as epilepsy. However, much is unknown about the mechanisms regulating specific neuropeptides to mediate behavior. Here, we report that the expression levels of dozens of neuropeptides are up-regulated in response to circuit activity imbalance in *C. elegans*. *acr-2* encodes a homolog of human nicotinic receptors, and functions in the cholinergic motoneurons. A hyperactive mutation, *acr-2(gf)*, causes an activity imbalance in the motor circuit. We performed cell-type specific transcriptomic analysis and identified genes differentially expressed in *acr-2(gf)*, compared to wild type. The most over-represented class of genes are neuropeptides, with insulin-like-peptides (ILPs) the most affected. Moreover, up-regulation of neuropeptides occurs in motoneurons, as well as sensory neurons. In particular, the induced expression of the ILP *ins-29* occurs in the BAG neurons, which were previously shown to function in gas-sensing. We also show that this up-regulation of *ins-29* in *acr-2(gf)* animals is activity-dependent. Our genetic and molecular analyses support cooperative effects for ILPs and other neuropeptides in promoting motor circuit activity in the *acr-2(gf)* background. Together, this data reveals that a major transcriptional response to motor circuit dysregulation is in up-regulation of multiple neuropeptides, and suggests that BAG sensory neurons can respond to intrinsic activity states to feedback on the motor circuit.

## Introduction

Neural circuits are dynamic, changing their properties in response to experience. These changes are critical for maintaining circuit homeostasis and in processes like memory. Many factors such as c-Fos and BDNF, are activated early in response to increased neural activity and further regulate the expression of downstream genes [1]. These early-acting genes are also involved in many neurological diseases. For example, mutations in the activity-dependent transcriptional repressor gene Mecp2 are associated with Rett's syndrome [2]. Mecp2 is

**Data Availability Statement:** Sequencing datasets have been deposited in the Gene Expression Omnibus (Accession GSE139212).

**Funding:** K.A.M. was a trainee on National Institutes of Health institutional training grants

(T32 NS007220 and T32 AG000216). This work was supported by a National Institutes of Health grant to Y. J. (R37 NS035546). The funders had no role in study design, data collection and analysis, decision to publish, or preparation of the manuscript.

**Competing interests:** The authors have declared no competing interests.

necessary for the transcriptional up-regulation of BDNF, a key early-acting gene regulating synaptic plasticity [3].

Neuropeptides are small, secreted molecules that play neuro-modulatory roles in all animals. Neuropeptides have an extraordinarily diverse set of functions, including in feeding, mood, and reproduction, among others. Secreted neuropeptides bind to G-protein coupled receptors (GPCRs) on target cells to modulate neuronal activity. Neuropeptides can act on cells post-synaptic to where they are secreted from, but can also act over long distances. Neuropeptide expression can be changed by experience, for example, the expression of Neuropeptide Y(NPY) changes in response to a myriad of stressors. NPY can inhibit anxiety in multiple stress models [4], and may also play a role in neurological diseases, such as epilepsy [5]. Although multiple neuropeptides have been implicated in diseases, much remains unknown about how they are regulated [5,6].

The nematode *C. elegans* has long been an important experimental model for investigating the regulation of neuronal circuits. The well-defined connectomics of its nervous system, in combination with powerful genetics and molecular tools, enable *in vivo* dissection of neural circuit regulation with high resolution. *C. elegans* locomotion is controlled through the balanced activities of cholinergic excitatory neurons and GABAergic inhibitory neurons to promote contraction and relaxation of body-wall muscle, respectively [7]. The locomotor circuit has been used to identify multiple conserved genes that regulate synaptic transmission [8,9]. The gene *acr-2* encodes a neuronal acetylcholine receptor subunit that is expressed in cholinergic motoneurons. A Valine-to-Methionine mutation causes a gain-of-function in *acr-2* [*acr-2 (gf)*] that results in a hyperactive channel [10]. The *acr-2(gf)* mutation affects a highly conserved residue within the pore-lining TM2 domain, and similar mutations in the human CHRNB2 cholinergic receptor subunit are associated with Autosomal Dominant Frontal Lobe Epilepsy (ADFLE) [11]. *acr-2(gf)* worms show uncoordinated (or Unc) movement as well as spontaneous whole-body shrinking, or convulsion.

Over 100 neuropeptide genes have been identified in *C. elegans*. These genes produce neuropeptides that fall into three classes: FMRFamide-like peptides (FLP), neuropeptide-like proteins (NLP) and insulin-like peptides (ILP) [12]. As with neuropeptides in humans, each neuropeptide gene produces a pro-neuropeptide, that is subjected to several enzymatic processing steps. The *flp* and *nlp* genes can generate several neuropeptides from a single locus through enzymatic cleavage by the pro-protein convertase *egl-3* and the endopeptidase *egl-21* [13,14]. Similar to human insulin and insulin-like growth factors, the *ins* loci produce a single peptide, that is activated from the pro-insulin peptide through the enzymatic activity of *egl-3* and a related pro-protein convertase *kpc-1* [15].

Previous work has shown that the neuropeptides *flp-18* and *flp-1* are important for regulating neurotransmission in *acr-2(gf)* mutants [16]. Loss of both *flp-18* and *flp-1* resulted in enhancement of *acr-2(gf)* motor defects. Additionally, *flp-18* expression was up-regulated in cholinergic motoneurons to inhibit convulsion of *acr-2(gf)* animals in a homeostatic manner. However, loss of function in the gene *unc-31*/CAPS, which is required for neuropeptide secretion, suppressed *acr-2(gf)* convulsion, suggesting that other neuropeptides function to promote circuit hyperactivity [17]. Using a cell-type specific transcriptomic approach, we identified over 200 genes whose expression was significantly altered in *acr-2(gf)* neurons compared to wild type. Among them, genes involved in neuropeptide signaling were significantly over-represented in this gene list. One of these, *ins-29*, has not been previously characterized. Expression reporters for *ins-29* are weakly or not expressed in BAG gas-sensing neurons in wild type, and *ins-29* expression is clearly increased in *acr-2(gf)* animals. The increased *ins-29* expression in BAG neurons of *acr-2(gf)* adults is activity-dependent and requires the transcription factor *ets-5*, which has been shown previously to act embryonically to make BAG functionally

competent [18]. Although BAG neurons interact with the motor circuit to respond to environmental cues, our data indicates that intrinsic activity states modulate expression of genes in the sensory BAG neuron, which then feed-back on the motor circuit. Finally, we present functional evidence supporting that the concerted action of several neuropeptides underlies motor circuit hyperactivity.

## Results

### Expression profiling of adult cholinergic neurons

We performed cell-type specific RNA-seq analyses, using the *Pacr-2*::*gfp* reporter *juIs14* to isolate GFP expressing neuronal cells by FACS followed by RNA-seq (Materials and methods, S1 Table) [10,19,20,21]. Besides expression in the ventral cord cholinergic motoneurons (VA, VB, DA, and DB), GFP expressed from *juIs14* is also present in several unidentified neurons in the head and tail [10]. RNA-seq data from isolated wild-type neurons was analyzed with the Cufflinks program to identify expressed genes (Materials and methods, S2 Table). The gene list for the neurons labeled by *Pacr-2*::*gfp* from wild type animals was compared with a recent study that profiled major tissue-types in adult *C. elegans* [22]. We found that almost all of the genes identified in our dataset (~95%) were also detected in a pan-neuronal analyses of expressed genes (Fig 1A), but shared very little overlap with tissue-specific expression profiles identifying enriched transcripts in hypodermis and muscle (Fig 1A). This comparison suggests our samples were relatively free of contamination from surrounding non-neuronal tissue. We also compared our gene list to previous analysis of a subset of cholinergic motoneurons (VA and DA) labeled with *Punc-4*::*gfp* [23]. Although the cell-types analyzed in these studies are not identical (A-type only vs. neurons expressing *Pacr-2*::*gfp*), and they were performed at different life stages (larval vs. young adult in this study) using different techniques (microarray vs. RNA-seq in this study), over half of the genes (~75%) from our dataset were also identified by microarray in larval type A motoneurons (Fig 1B). These include core cholinergic genes such as *unc-17*/VAChT (Avg. FPKM = 484.1) and pan-neuronal genes such as *unc-13* (Avg. FPKM = 90.9), required for synaptic vesicle priming [24,25]. Additionally, both of these datasets were relatively free of transcripts enriched in GABAergic motoneurons. Thus, this comparison shows the genes identified in our sample are highly enriched for those that are validated to be expressed and function in cholinergic neurons.

### Differential expression analyses between wild-type and *acr-2(gf)* neurons

We are interested in the genes differentially expressed in response to altered neuronal activity. The transcriptome of neurons labeled by *Pacr-2*::*gfp* in adult *acr-2(gf)* mutants was compared to those from wild type animals for changes in gene expression using DESeq2. This analysis identified 234 genes as significantly mis-expressed in *acr-2(gf)* mutants (S2 Table) [26,27]. We found *flp-18* to be significantly up-regulated in *acr-2(gf)*, as predicated from our previous studies [16]. Analysis of the distribution of the fold change data using a histogram showed that a majority of the changes were up-regulation, at around 2-fold (Fig 1C). GO-term analyses [28] indicated that genes involved in neuropeptide signaling were highly enriched in our gene list (Fig 1D). In total, the expression of 21 neuropeptide genes were identified as significantly up-regulated in *acr-2(gf)* neurons, representing approximately 18% of the estimated 113 neuropeptide genes identified in *C. elegans* (Fig 2A) [12]. Of these, several insulin-like peptides (ILP) were the most up-regulated (Fig 2A, S2 Table). In addition, none of these ILP genes were detected as expressed in wild type samples (S2 Table). Together, these data indicate that the major response to altered motor circuit activity at the transcriptional level is to increase neuropeptide gene expression.

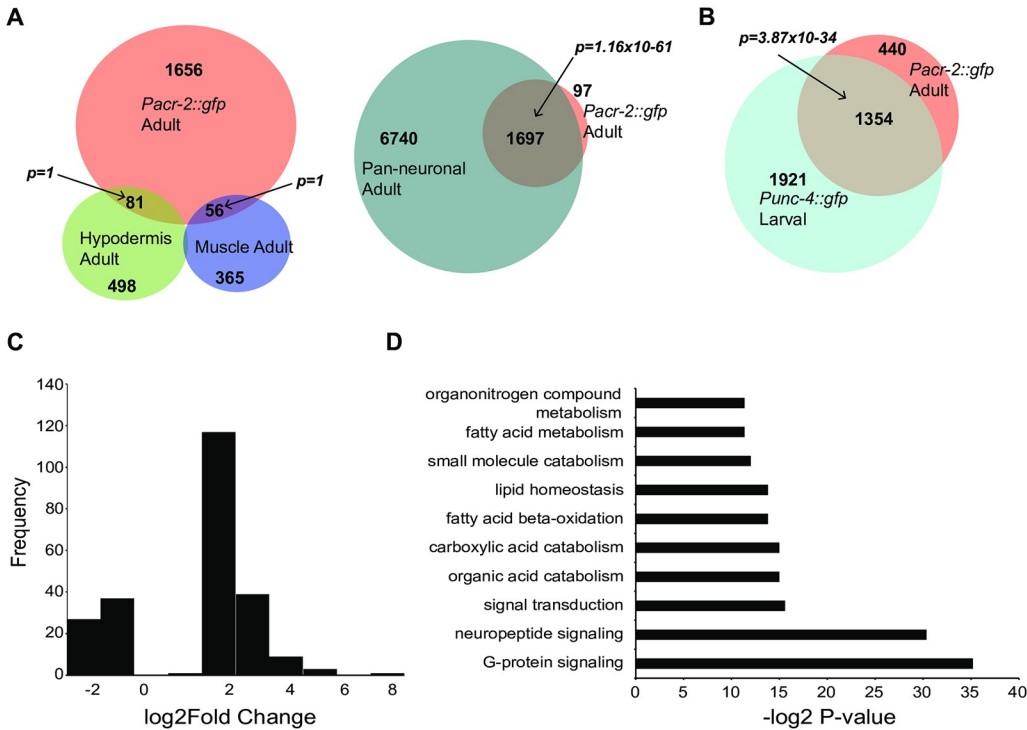

**Fig 1. Differential expression analyses of *acr-2(gf)* neurons compared to wild type.** A. Few genes identified as expressed in isolated adult neurons labeled by *Pacr-2::gfp* are enriched in hypodermal or muscle datasets [22]. In contrast, almost all of the genes identified as expressed in cholinergic neurons labeled by *Pacr-2::gfp* are identified in pan-neuronal data-sets. Shown are Venn Diagrams with overlaps between the indicated data sets. The number in each region indicates the number of genes in that category. Hypodermis, muscle, and neuronal datasets are from Kaletsky *et al.* (2018). Statistics for significance of overlap were performed using a hypergeometric distribution with the *phyper* function in R. B. Genes identified in previous expression profiling of larval cholinergic motoneurons, including core neuronal and cholinergic markers, are identified by RNA-seq of *Pacr-2::gfp* expressing neurons [23]. The Venn Diagram displays that over half of genes identified by microarray as expressed in larval A-type motoneurons are also identified by RNA-seq in adult cholinergic neurons that express P*acr-2::gfp*. The number in each region indicates the number of genes in that category. Larval A-type motor neuron expression data is from Von Stetina *et al.* (2007). Statistics for significance of overlap were performed using a hypergeometric distribution with the *phyper* function in R. C. Histogram of log$_2$(Fold Change) for significantly different genes. Most genes that were different in *acr-2(gf)* compared to wild type were up-regulated at around 2-fold. D. GO-term analyses of genes significantly different in *acr-2(gf)* neurons compared to wild type (see Materials and methods). Genes involved in neuropeptide and G-protein signaling (which included neuropeptide genes) were the most affected.

To validate the RNA-seq analyses, we analyzed transcriptional reporters for the most up-regulated neuropeptide gene of each class. *flp-12* and *nlp-1* were the most up-regulated neuro-peptides of their respective classes. These genes have been implicated in context-dependent behaviors. *flp-12* is involved in male locomotion, and *nlp-1* functions in food-evoked turning behaviors [29,30]. 2kb upstream sequences of *flp-12* and *nlp-1* were used to generate GFP reporters, and both reporters showed expression in neurons in the head, similar to published studies of these neuropeptides (Fig 2B–2E) [31,32]. In *acr-2(gf)*, the *flp-12* reporter showed a different expression pattern than wild type, with ectopic expression in a pair of neurons close to the ganglion (Fig 2B and 2D). Similarly, the *nlp-1* reporter was up-regulated and expressed in additional cells in *acr-2(gf)* animals compared to wild type (Fig 2C and 2E). Therefore, this analysis provides an explanation for the increased RNA levels detected in RNA-seq of *acr-2 (gf)*, and also shows that increased locomotor excitation may cause mis-expression of these neuropeptide genes.

**A**

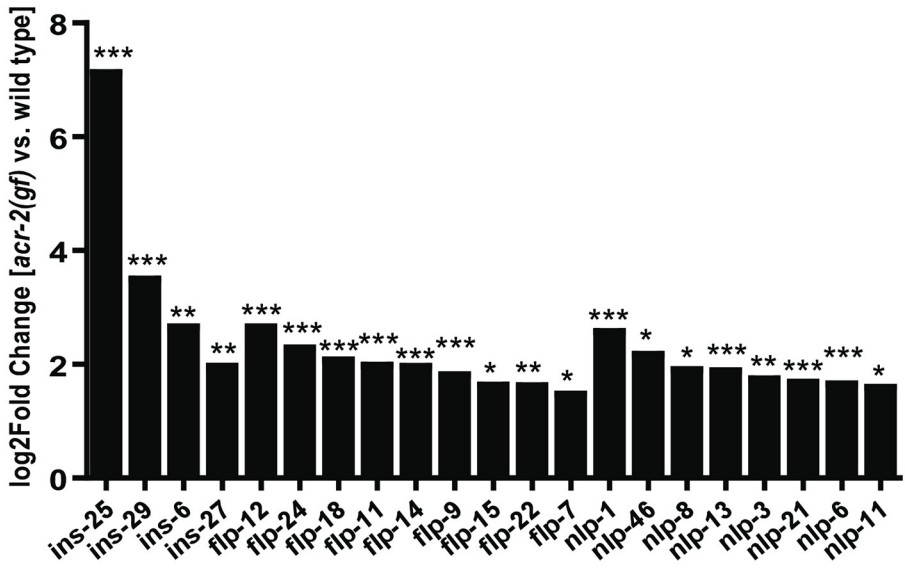

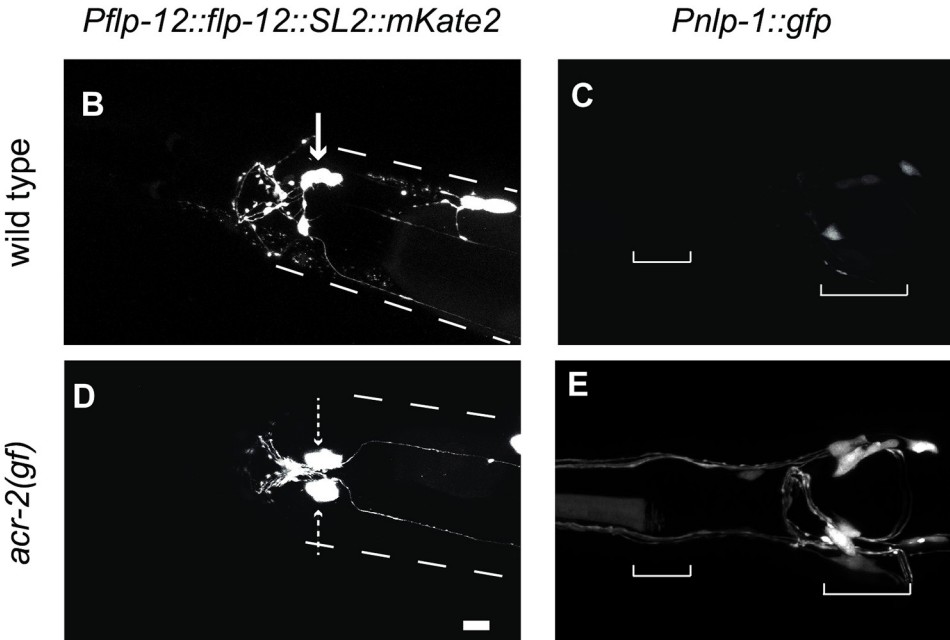

**Fig 2. Neuropeptides are up-regulated in head neurons in *acr-2(gf)* animals.** A. $Log_2$(Fold Change) values for all up-regulated neuropeptides in *acr-2(gf)* animals are shown, grouped by class, as identified with DESeq2 analyses. Neuropeptides of each class were affected. (*$P<0.05$, **$P<0.01$, ***$P<0.001$). B-E. Differential expression of neuropeptide transcriptional reporters in the head of *acr-2(gf)* animals compared to wild type. (B,D) The arrow points to the SMB neuron expressing the P*flp-12* transcriptional reporter (*juEx7964*) in wild type based on shape and location. Dashed arrows point to the bilateral pair ectopically expressing the reporter in *acr-2(gf)*. Dashed lines indicated approximate outline of the animal. The posterior fluorescent signal in both strains is the co-injection marker labeling coelomocytes.(C,E) P*nlp-1::gfp (juEx7879)* expression is both increased in the same cells as wild type and also ectopically expressed. Brackets delineate the anterior and posterior pharyngeal bulbs, respectively.

## *acr-2(gf)* increases expression of *ins-29* and *acr-2* in BAG sensory neurons

Among all up-regulated neuropeptides, *ins-25* and *ins-29* showed the most dramatic increase (Fig 2A). As the expression pattern and function of these ILPs is unknown, we investigated their expression patterns in further detail. *ins-25* and *ins-29* are in an operon on Chromosome I, with *ins-29* being upstream of *ins-25*, separated by 697bp intergenic sequence (Materials and methods, Fig 3A). We used 2kb upstream sequences of *ins-29* to drive GFP expression. In wild type animals, the *Pins-29* transcription reporter was sometimes weakly expressed in two neurons in the head, but often expressed in just one neuron or no GFP expression was detected (Fig 3B). However, in *acr-2(gf)* animals, consistent strong expression of GFP was observed in the same two head neurons (Fig 3C). These neurons are likely sensory, as they extend dendrites out to the nose of the animal.

To confirm that the *ins-29* expression construct was expressed in cells labeled by *Pacr-2*::*gfp*, we generated an *ins-29* expression construct containing the endogenous *ins-29* transspliced to mKate2 (Fig 3D–3I). The *Pins-29*::*ins-29*::*SL2*::*mKate2* expression pattern was similar to *Pins-29*::*gfp* (Fig 3B–3D and 3G). We then generated strains co-expressing *Pins-29*::*ins-29*::*SL2*::*mKate2* and *Pacr-2*::*gfp*. In wild-type, *Pacr-2*::*gfp* was expressed in several unidentified neurons in the head, in addition to its reported expression in ventral cord cholinergic motoneurons [10]. No co-localization was observed between *Pacr-2*::*gfp* and the *Pins-29*::*ins-29*::*SL2*::*mKate2* in wild type animals that showed expression of the *ins-29* reporter (Fig 3D–3F). However, in *acr-2(gf)* animals, the expression of *Pins-29*::*ins-29*::*SL2*::*mKate2* showed co-localization with *Pacr-2*::*gfp* (Fig 3G–3I). This result suggests that, although total mRNA levels of *acr-2* in neurons are not significantly affected by the *acr-2(gf)* mutation, there is increased expression of *acr-2* in some head neurons compared to wild type.

The dendrite morphology and cell position of cells expressing *ins-29* suggested that they were likely to be the BAG neurons, which are known for their role in gas-sensing, particularly $CO_2$ avoidance (Fig 3C) [33,34,35]. To confirm that the cells expressing *ins-29* were indeed BAG neurons, animals were generated co-expressing the *Pins-29*::*ins-29*::*SL2*::*mKate* reporter and an integrated GFP marker for BAG [*Pgcy-33*::*gfp*] (Fig 4A–4F). We observed that in *acr-2 (gf)* animals, where the *ins-29* reporter expression was consistently detected, the expression of mKate2 completely overlapped with GFP (Fig 4D–4F). For those wild type animals that expressed the *ins-29* reporter, mKate2 also overlapped with the BAG GFP marker (Fig 4A–4C).

The transcription factor *ets-5* is required for BAG development and function [18]. To determine if *ets-5* is involved in the expression of *ins-29* in *acr-2(gf)*, an *ets-5(0) acr-2(gf)* double mutant strain with *Pins-29*::*gfp* was generated. Penetrance of expression was quantified by scoring for presence of GFP expression in zero, one, or both BAG neurons (Fig 4G). Very few wild-type animals expressed the transgene in both BAG neurons (31%), while most *acr-2(gf)* animals (99%) did. We found a marked decrease in the expression of the *Pins-29*::*gfp* reporter in *ets-5(0) acr-2(gf)* mutants, similar to wild type. Altogether, these data indicate that in *acr-2 (gf)* animals, *ins-29* is up-regulated in sensory BAG neurons in an *ets-5*-dependent manner.

## *ins-29* expression in BAG neurons is regulated by systemic *acr-2(gf)* activity

To address whether the up-regulation of *Pins-29*::*gfp* in response to *acr-2(gf)* is dependent on neuronal activity, we next tested if reduction in motor circuit hyperactivity could restore expression of *ins-29* to wild type. The TRPM channel *gtl-2* is expressed in non-neuronal tissues, such as the hypodermis, and regulates motor circuit hyperexcitation via systemic ion homeostasis [36]. Null mutation of *gtl-2* almost completely suppresses *acr-2(gf)* convulsion and locomotion phenotypes. The expression of *Pins-29*::*gfp* was analyzed in both *gtl-2(0)* and

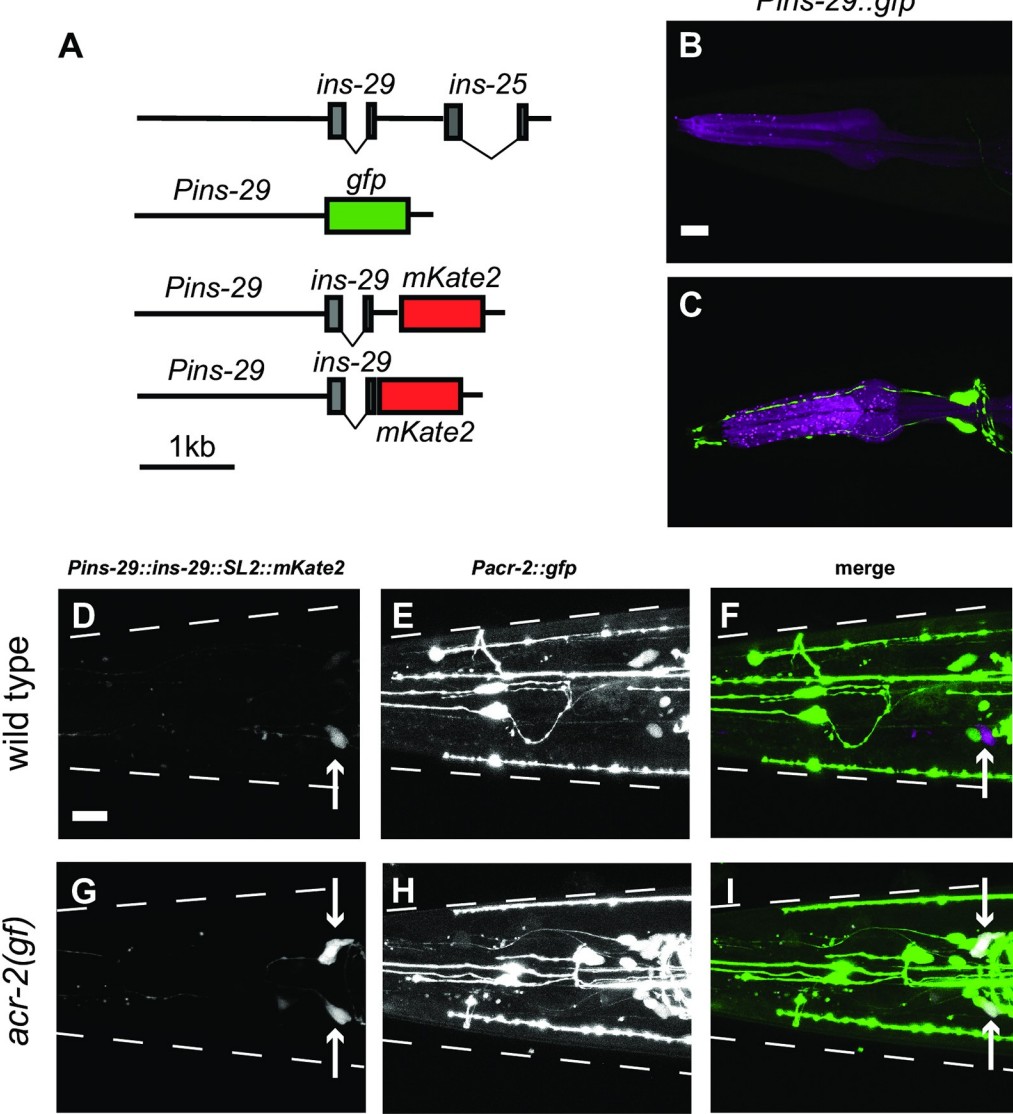

**Fig 3. Expression of ILP genes and *acr-2(gf)* is up-regulated in two head neurons in *acr-2(gf)* animals.** A. Shown is the gene structure of *ins-29* and *ins-25* loci. Also shown are diagrams of the *ins-29* transcription reporters and expression constructs used in this study (Listed in S1 Table). Two *ins-29* expression constructs were made, one that contained *SL2::mKate2* sequence to monitor transgene expression, the other has mKate2 in-frame fused to INS-29, without the SL2 trans-splice sequence. B-C. *Pins-29::gfp* is expressed in two head neurons in *acr-2(gf)*. scale bar:10μm. (B) in wild-type *Pins-29::gfp(juEx7742)* is often detected in one or zero cells in the head. (C) In *acr-2(gf)*, *Pins-29::gfp* is strongly expressed in two head neurons. D-I. A *Pins-29::ins-29::SL2*::mKate2 reporter*(juEx7966)* co-localizes with *Pacr-2::gfp(juIs14)* expression in *acr-2(gf)* animals, but not in wild type. scale bar:10μm. Cell bodies expressing mKate2 are labeled by an arrow. (D) In wild type animals, *Pins-29::ins-29::SL2*::mKate2*(juEx7966)* is weakly or not expressed in the head. Shown is an animal expressing the transgene in a single neuron. (E) *Pacr-2::gfp* is expressed in multiple neurons in the head of wild type animals. (F) Expression of *ins-29* and *acr-2* transcriptional reporters do not overlap in wild type animals, suggesting that *acr-2* is not normally expressed in the same neurons as *ins-29* in wild type. (G) Expression of *Pins-29::ins-29::SL2*::mKate2 in *acr-2(gf)* is similar to that of *Pins-29::gfp* observed in (C). (H) Expression of *Pacr-2::gfp* reporter in *acr-2(gf)* animals. (I) Co-localization is observed between the *ins-29* reporter and *Pacr-2::gfp* in *acr-2(gf)* animals. In all images, animals are rolled ventral-side up for easy visualization of neuron pairs. The dashed lines represent the approximate outline of the animal. The two processes in the head of animals expressing the *Pins-29::ins-29::SL2::mKate2* are mechanosensory neuron dendrites labeled by the *Pmec-4::gfp* co-injection marker. Beading observed in some images (i.e. the *Pacr-2::gfp* in E, H) is due to rolling the animals in 10% agarose.

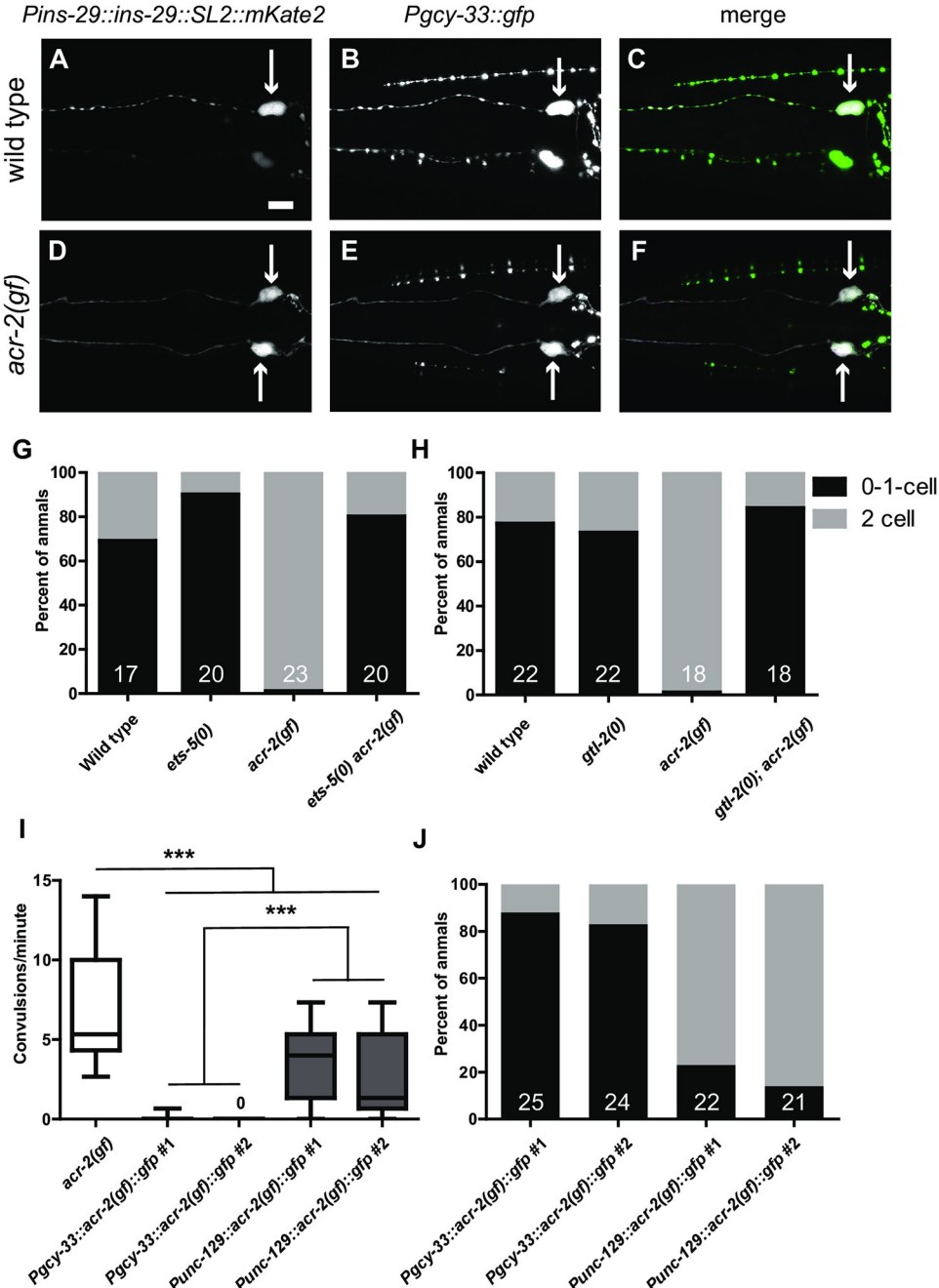

**Fig 4. *ins-29* is expressed in BAG neurons and is regulated by motor circuit activity.** A-F. *Pins-29* reporter is expressed in BAG neurons. Cell bodies expressing indicated reporters are labeled by an arrow. (A) In wild type animals, expression of *Pins-29::ins-29::SL2::mKate2* in a single neuron in the head is shown. This reporter is often not expressed at all or in a single neuron in wild type, similar to the transcriptional GFP reporter (B) *Pgcy-33::gfp* labels BAG neurons in the head. (C) Overlap between the two reporters can be observed in one neuron in wild type. (D) Expression of *Pins-29::ins-29::SL2::mKate2* is observed in two head neurons in *acr-2(gf)*, similar to the transcriptional GFP reporter. (E) Expression of *Pgcy-33::gfp* in *acr-2(gf)* is similar to wild type in *acr-2(gf)*. (F) Complete overlap is observed between the two reporters in *acr-2(gf)*. GFP-only signal is from the *Pmec-4::gfp* co-injection marker. Scale bar:10μm. In all images, animals are rolled in 10% agarose to facilitate visualization of BAG neuron pairs. Beading observed in some images (i.e. the *Pgcy-33::gfp* and *Pmec-4::gfp* in B) is due to the mounting procedure. G. Almost all *acr-2(gf)* animals express *Pins-29::gfp(juEx7742)* in both BAG neurons, however mutation of *ets-5* causes animals to exhibit more similar *Pins-29::gfp* expression patterns as wild type. To assay *Pins-29::gfp* expression, animals were scored for detectable GFP in 0, 1, or 2 BAG neurons. N for each genotype is labeled in the bar. H. Loss-of-function of

the TRPM channel *gtl-2*, which almost completely suppresses *acr-2(gf)* convulsion and locomotion phenotypes, also restores *Pins-29::gfp(juEx7742)* expression to wild-type patterns. To assay *Pins-29::gfp* expression, animals were scored for detectable GFP in 0, 1, or 2 BAG neurons. N for each genotype is labeled in the bar. I. Expression of *acr-2(gf)* in BAG neurons does not cause convulsions. Shown are convulsion rates of *acr-2(gf)* animals and transgenic animals expressing cell-type specific *acr-2(gf)*. *Pgcy-33::acr-2(gf)::gfp* (*juEx8070*, *juEx8071*, arrays also contain *Pins-29::gfp* sequence) does not result in convulsion. *Punc-129::acr-2(gf)::gfp* (*juEx8068*, *juEx8069*) induces convulsion, albeit at a lower frequency than in *acr-2(gf)* mutants. Box-and-whisker plots are shown with the middle bar representing the median. ***P<0.001, One-way ANOVA followed by Bonferroni post-hoc test. N≥13 for each genotype. J. Expression of *acr-2(gf)* in a subset of motor neurons, but not BAG neurons, causes almost complete penetrance of *Pins-29::gfp* expression in BAG neurons. To assay *Pins-29::gfp* expression, animals were scored for detectable free GFP in cell body and dendrites in 0, 1, or 2 BAG neurons. Animals carrying *Pgcy-33::acr-2(gf)::gfp* arrays (*juEx8070*, *juEx8071*, arrays also contain *Pins-29::gfp* sequence) or *Punc-129::acr-2(gf)::gfp* (*juEx8068*, *juEx8069*, arrays also contain *Pins-29::gfp* sequence) were scored. N for each genotype is labeled in the bar.

*gtl-2(0); acr-2(gf)* backgrounds. Animals were scored for GFP expression in zero, one, or both neurons (Fig 4H). *gtl-2(0)* animals alone resembled wild type. The expression pattern of *Pins-29::gfp* in *gtl-2(0); acr-2(gf)* animals also resembled wild type. This result supports the idea that the changes in neuropeptide expression are likely due to systemic motor activity.

In light of the finding that *acr-2* expression was up-regulated in BAG neurons (Fig 3D–3I), we wished to discriminate whether *ins-29* expression was regulated by *acr-2(gf)* activity in BAG neurons or the motor circuit. We expressed *acr-2(gf)* cDNA in a subset of motor neurons or in BAG neurons and assessed *Pins-29::gfp* expression. As previously reported, expression of *acr-2(gf)* using the *unc-129* promoter (which drives expression in the ventral cord DA and DB motoneurons) caused animals to display an Unc phenotype as well as convulsions, albeit at a lower frequency than *acr-2(gf)* mutants (Fig 4I) [37]. In contrast, BAG neuron-specific expression of *acr-2(gf)* driven by the *gcy-33* promoter did not cause convulsion. We quantified penetrance of *Pins-29::gfp* reporter expression in animals expressing *acr-2(gf)* in cholinergic motoneurons or BAG neurons and found that almost all animals expressing *acr-2(gf)* in this small set of cholinergic motoneurons showed expression of the *Pins-29::gfp* reporter in both BAG neurons, similar to *acr-2(gf)* mutant animals (Fig 4J). However, animals with the *Pgcy-33::acr-2(gf)* transgenes had much lower penetrance of *Pins-29* reporter expression. Taken together, this data support that increasing motor circuit activity results in a cell non-autonomous effect on *ins-29* expression in BAG neurons.

Neuropeptides are typically secreted from the cell in which they are produced and can act at a distance (endocrine signaling) as well as on post-synaptic or nearby cells (paracrine signaling). We next sought to address whether INS-29 can be secreted, by generating a transgene expressing INS-29::mKate2 fusion protein. Secreted molecules are present in the coelom (or interstitial fluid), which are generally taken up by the coelomocytes. We found mKate2 in the vacuoles of posterior coelomocytes in both wild type and *acr-2(gf)* animals (S1 Fig). This observation shows that INS-29 is secreted and may therefore act on multiple cells and tissues within the animal.

## Insulin-like peptides and *flp-12* coordinately promote motor circuit activity

Next, we addressed whether these neuropeptides expressed from head neurons affect motor circuit function. Genetic null [designated as *(0)*] mutations were used to determine if candidate up-regulated neuropeptides had functional roles in the *acr-2(gf)* motor circuit (Materials and methods, Fig 5A). We focused on the most up-regulated neuropeptide genes of each class. Single loss of function mutations of *ins-25*, *ins-29*, *flp-12*, or *nlp-1* did not significantly affect convulsion rate of *acr-2(gf)* (Fig 5B and 5C). Additionally, a large deletion mutation, *ju1580*,

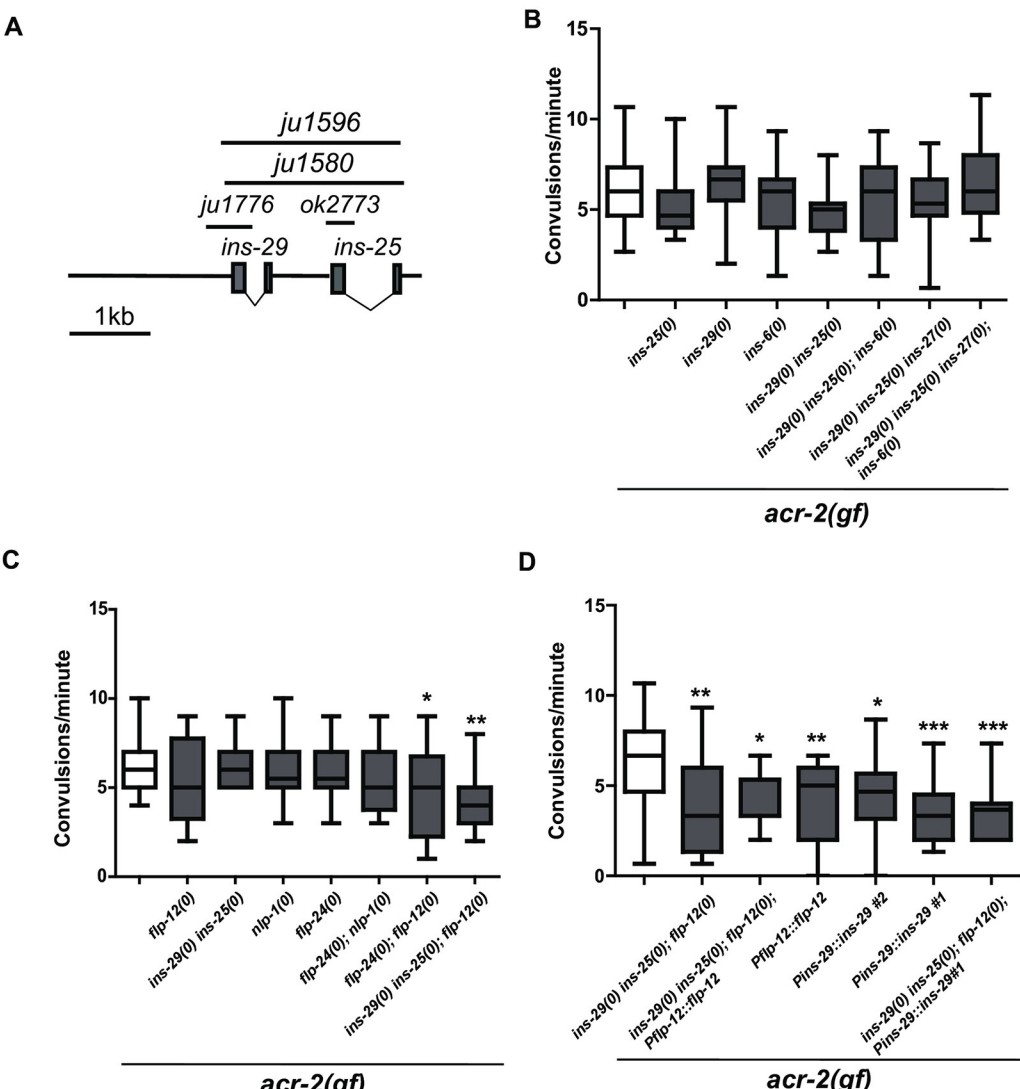

**Fig 5. ILPs and *flp-12* coordinately promote motor circuit activity.** A. Shown is the *ins-29* and *ins-25* gene structure with illustrations of the deletion mutants used in this study. *ok2773* is a 354bp deletion in *ins-25*. *ju1776* is a 568bp deletion in *ins-29*. *ju1580* is a 2.2kb deletion of both *ins-25* and *ins-29*, and a similar deletion, *ju1596*, is made in *ins-27 (ok2474)* genetic background, which is ~6kb 3' to *ins-25*. B. Convulsion rate of *acr-2(gf)* combined with different combinations of deletions in genes for insulin-like peptides up-regulated in *acr-2(gf)* neurons is shown as convulsions/minute. None of these combinations had a statistically significant effect of convulsion rate. Box-and-whisker plots are shown with the middle bar representing the median (One-way ANOVA followed by Dunnett's test. N≥18). C. Convulsion rates for indicated genotypes are shown. Deletion of *flp-12* in combination with either *flp-24(0)* or deletion of both *ins-29* and *ins-25* causes a statistically significant reduction in convulsion rate. Box-and-whisker plots are shown with the middle bar representing the median (*P<0.05, **P<0.01. One-way ANOVA followed by Dunnett's test. N≥19). D. Over-expression of either *flp-12* or *ins-29* suppresses convulsion. Over-expression arrays of *flp-12(juEx7964)* or *ins-29 (juEx7966, juEx7967)* were crossed into indicated mutant backgrounds and scored for convulsion. Box-and-whisker plots are shown with the middle bar representing the median (*P<0.05, **P<0.01, ***P<0.001. One-way ANOVA followed by Dunnett's test. N≥19).

that removed both *ins-29* and *ins-25* did not significantly affect convulsion rate (Fig 5B and 5C, designated as *ins-29(0) ins-25(0)*). However, a slight decrease in convulsion rate was detected in *flp-12(0) acr-2(gf)* animals (Fig 5C). Therefore, several combinations were made between *flp-12(0) acr-2(gf)* animals and null alleles of *ins-29 ins-25* or *flp-24*, the second most

up-regulated *flp* gene. Analysis of convulsion rates in these strains showed that deletion of *flp-12* with either *flp-24* or *ins-29 ins-25* resulted in a significant decrease in convulsion rate compared to *acr-2(gf)* alone (Fig 5C). This analysis suggests that these neuropeptides act in a combinatorial manner to promote circuit hyperactivity.

We also sought to restore convulsion frequency in *ins-29(0) ins-25(0); flp-12(0) acr-2(gf)* compound mutants by over-expressing wild type *ins-29* or *flp-12* in *ins-29(0) ins-25(0); flp-12 (0) acr-2(gf)* mutant animals. Interestingly, over-expression of either *ins-29* or *flp-12* alone in *acr-2(gf)* animals also partially suppressed convulsion frequency to a similar degree as the neuropeptide compound mutant background (Fig 5D). This observation implies that over-expression of either *ins-29* or *flp-12* can act in a dominant and interfering manner to suppress *acr-2 (gf)* convulsion.

We further asked whether these neuropeptide genes affected synaptic transmission in the locomotor circuit using pharmacological assays. Aldicarb is an acetylcholinesterase inhibitor which leads to buildup of acetylcholine at the synaptic cleft and eventual paralysis in wild type animals [38]. Mutants that increase or decrease cholinergic activity display increased or decreased sensitivity to aldicarb. *acr-2(gf)* animals are more sensitive to aldicarb, consistent with their hyperactivity as described by previous electrophysiology and pharmacology analysis [10]. *ins-29(0) ins-25(0); flp-12(0) acr-2(gf)* animals were similar to *acr-2(gf)* alone on aldicarb, indicating that overall neurotransmission was not strongly affected by these mutations (Fig 6A, S3 Table). *ins-29(0) ins-25(0); flp-12(0)* mutants were also not significantly different from wild type animals on aldicarb (S4 Table). Levamisole is an agonist for the post-synaptic cholinergic receptors on muscle [39,40]. Resistance or sensitivity to levamisole can represent changes in muscle responsiveness. In these experiments, we found that *ins-29(0) ins-25(0); flp-12(0) acr-2(gf)* animals were less sensitive to levamisole than *acr-2(gf)* alone (Fig 6B, S5 Table). However, *ins-29(0) ins-25(0); flp-12(0)* triple mutants showed similar levamisole sensitivity to wild type (S6 Table). This result suggests that these neuropeptides may function post-synaptically to affect motor circuit function, and their function is only observed in the *acr-2(gf)* background, consistent with these peptides being differentially expressed in response to *acr-2(gf)* hyperactivity.

We also asked if transgenic over-expression of *ins-29* could rescue the levamisole phenotype of *ins-29(0) ins-25(0); flp-12(0) acr-2(gf)* animals. Animals expressing the *Pins-29::ins-29::SL2:: mKate2* transgene in wild-type or mutant backgrounds were assessed for levamisole sensitivity. We observed that over-expression of *ins-29* in wild type exhibited a levamisole resistance phenotype, indicating that *ins-29* can inhibit post-synaptic activity in the absence of *acr-2(gf)* when over-expressed (Fig 6C, S7 Table). On the other hand, and in contrast to the convulsion analyses, *ins-29* over-expression did not affect the levamisole hypersensitivity of *acr-2(gf)* animals, and *acr-2(gf)* animals with or without the transgene had similar levamisole sensitivity profiles. *ins-29* over-expression, however, significantly rescued the suppression of *acr-2(gf)* by *ins-29(0) ins-25(0); flp-12(0)* mutations. These results of *ins-29* over-expression indicate that the function of these neuropeptides in motor circuit regulation may be context-dependent, with different activities depending on the genetic background and state of the circuit.

## Discussion

Using neuronal-type specific RNA-seq, we identified over 200 genes whose expression levels are altered in response to hyperactivity in the motor circuit of *C. elegans*. Genes encoding neuropeptides were over-represented in this list, and we validated that fluorescent reporters for *ins-29*, *flp-12*, and *nlp-1* were over- and/or ectopically-expressed in *acr-2(gf)* animals compared to wild type. These data support the conclusion that the major transcriptional response to

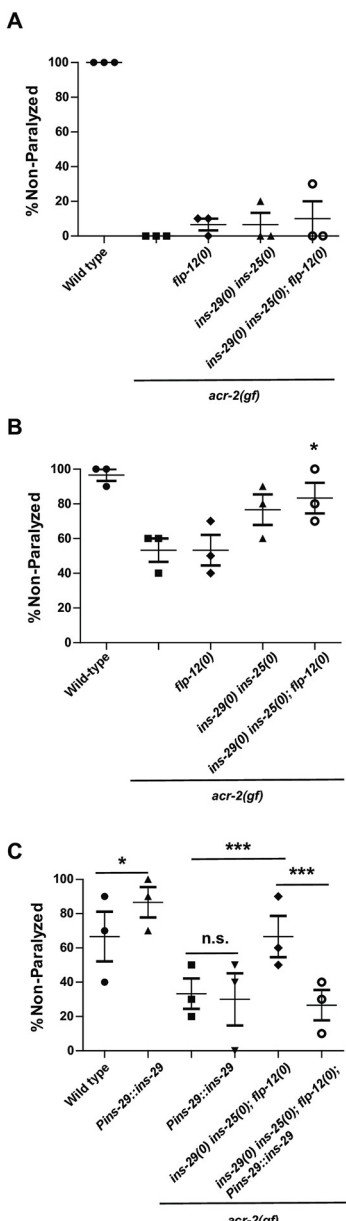

**Fig 6. ILPs regulate post-synaptic muscle excitability.** A. Aldicarb sensitivity shown as percentage of animals not paralyzed after 1hour on the drug. No significant difference was observed (Two-way ANOVA followed by Dunnett's test, compared to *acr-2(gf)* alone.). Each data point represents one trial and 10 animals per genotype were scored per trial. Data is also shown in S3 Table. B. Levamisole sensitivity of neuropeptide mutants in the *acr-2(gf)* background at 15min. Deletion of insulin-like peptides with *flp-12(0)* significantly reduces levamisole sensitivity compared to *acr-2 (gf)* at this timepoint. Each data point represents one trial and 10 animals per genotype were scored per trial (*P<0.05, Two-way ANOVA followed by Dunnett's test compared to *acr-2(gf)* single mutant.). Data is also shown in S5 Table. C. Effect of *ins-29* over-expression on levamisole sensitivity. Shown are a subset of mutant strains analyzed in Fig 6B also carrying the *ins-29* over-expression transgene (*juEx7966*). Over-expression of *ins-29* can restore *acr-2(gf)* levels of levamisole sensitivity to that of *ins-29(0) ins-25(0); acr-2(gf)* strains, but also inhibits levamisole sensitivity in wild-type circuits. Each data point represents one trial and 10 animals per genotype were scored per trial (***P<0.001, *P<0.05. Two-way ANOVA followed by Bonferroni test.). Data is also shown in S7 Table.

motor circuit hyperactivity in *C. elegans* is by altering neuropeptide gene expression. Bioinformatic analyses of upstream sequences did not identify common motifs in the promoter regions of theses neuropeptides, suggesting that multiple transcriptional pathways may be involved in these changes. Indeed, the fluorescent reporters analyzed here showed diverse expression patterns as well as varied changes in expression, from over-expression in the same cells as wild type, to ectopic expression patterns.

ILPs were the most up-regulated genes identified by RNA-seq in *acr-2(gf)*. We have previously shown that loss of function mutations in *unc-31*/CAPS, which is required for the release of the majority of neuropeptides, reduce convulsion rate of *acr-2(gf)* mutants, indicating that the release of unidentified neuropeptides modulates motor circuit hyperactivity [16,17]. Here, we show that loss of the ILP genes *ins-29* and *ins-25*, along with *flp-12*, can reduce convulsion of *acr-2(gf)* animals. This effect is modulatory, as mutation of these neuropeptides partially suppresses, but does not block, convulsion. The effects of these peptides are in contrast to our previous studies on a different neuropeptide *flp*-18, which is also up-regulated in cholinergic motoneurons of *acr-2(gf)* (this study), yet loss of *flp-18* function, together with loss of *flp-1*, increased convulsion of *acr-2(gf)* animals [16]. These data are consistent with many studies showing that neuropeptide modulation involves combinatorial and redundant action of different peptides that can have varied effects on epileptic states [5]. Paradoxically, we further find that overexpression of wild type *ins-29* or *flp-12* in *acr-2(gf)* reduced convulsion. We speculate that the observed effects from overexpressing each peptide reflect promiscuity of receptors for the neuropeptides. Indeed, the variety of effects observed by *ins-29* over-expression would be consistent with it affecting multiple pathways. Future studies will involve defining specific receptors for each peptide to clarify their function.

The expression and function of *ins-29* has not been previously characterized. Examination of *ins-29* transcriptional reporters showed expression in the gas-sensing BAG neurons, with increased intensity and more penetrant expression in both BAG neurons detected in *acr-2(gf)* compared to wild type. Furthermore, co-labeling also showed up-regulation of *acr-2* itself in these neurons. We show that the regulation of *ins-29* expression in BAG neurons is activity-dependent. For example, genetic suppression of *acr-2(gf)* by null mutation of *gtl-2* restored P*ins-29*::*gfp* expression to wild type patterns. BAG neurons are necessary for *C. elegans* to respond to changes in environmental $CO_2$ [33]. The observation that *ins-29* is expressed in the BAG sensory neurons, rather than motoneurons or pre-motor interneurons, was surprising. For example, whereas *flp-18* is expressed from the cholinergic motoneurons to affect *acr-2(gf)* activity and motor function, *ins-29* expression is increased in sensory neurons in the head and INS-29 is secreted. Together, these results indicate that genes involved in cholinergic neurotransmission including the *acr-2* cholinergic receptor subunit gene, as well as the insulin-like peptide gene *ins-29* are up-regulated in BAG neurons due to motor circuit hyperactivity in the *acr-2(gf)* background. Neuropeptide signaling from BAG has been shown previously to interact with another cholinergic circuit, the egg-laying circuit [41]. *flp-17* and *flp-10* secreted from BAG act in parallel with cholinergic signaling to inhibit egg-laying. It is proposed that signaling from BAG integrates favorable environmental signals with the egg-laying circuit.

The BAG-specific transcription factor *ets-5* also regulates P*ins-29*::*gfp* expression. Analysis of the *ins-29* promoter did not reveal a clear *ets-5* biding site, and it is possible this effect in indirect. *ets-5*-dependent pathways, therefore, are necessary for the increased expression of P*ins-29*::*gfp* in response to *acr-2(gf)*. This result also indicates a function for *ets-5* beyond development and maintenance of BAG neuron identity, but in modifying transcription in the BAG neurons under different physiological conditions in mature animals.

In humans, seizure is a common co-morbidity of diabetes. *In vitro* analysis has also found, for example, that IGF signaling can be neuroprotective in response to injury, but can also

promote epileptogenesis [42]. Insulin peptide signaling may also play a role in Alzheimer's Disease (AD) [6]. Although the role for insulin signaling in the brain is still unclear, levels of insulin and insulin receptors are markedly decreased in the brains of AD patients. Our data show that in *C. elegans*, ILP signaling is altered by aberrant cholinergic activity to modulate circuit function.

## Materials & methods

### *C. elegans* genetics

*C. elegans* strains were maintained at 20–22˚C. For RNA-seq experiments, CZ631(*juIs14[Pacr-2::gfp]*) and CZ5808 (*juIs14[Pacr-2::gfp]; acr-2(n2420gf)*) were used. S1 Table contains a list of all strains. All genetic null alleles are designated by *(0)* in the text.

CRISPR mutagenesis was performed as described previously, via co-CRISPR with *dpy-10* marker [43]. Two sgRNAs were designed to bind outside the *ins-29* and *ins-25* region [*ins-29* TTGGCGCCCAGCGCCGTTGT GGG, *ins-25* CAGATCTTCGATTGGGACGG CGG]. To make the *ins-29(ju1776)* single deletion, the 5' *ins-29* sgRNA was injected, and CRISPR editing resulted in a 568bp deletion spanning the entire first exon of *ins-29*. Both sgRNAs were injected into wild-type or *ins-27(ok2474)* animals, and a ~2.2kb mutation that deleted both *ins-29* and *ins-25* was obtained, resulting in an *ins-29(0) ins-25(0)* double mutant (*ju1580*) or *ins-29(0) ins-25(0) ins-27(0)* triple mutant (*ju1596 ok2474*), respectively. These compound mutations were outcrossed, and then introduced into *acr-2(gf)*. Most crosses (except those described below) were done using standard methods. See S8 Table for genotyping primers.

Construction of double mutant strains with *flp-12(ok2309)* or *ets-5(tm866)* with *acr-2(gf)* was made using the MT6448 *lon-2(e678) acr-2(gf)* strain. *flp-12* (X:-7.20) and *ets-5* (X:-6.20) are to the left of *acr-2* (X:-2.56) on the X chromosome. We generated heterozygotes of the genotype: *lon-2(e678) acr-2(gf)* X/[*flp-12(ok2309)* or *ets-5(tm866)*] X. Non-long, convulsing progeny were isolated from the next generation and genotyped for the gene of interest.

### Sample preparation for RNA-seq

Samples were prepared for dissociation and FACS essentially as described [19,44]. Animals were synchronized by hypochlorite treatment. Eggs were isolated from wild type N2, CZ631, and CZ5808 gravid adults and allowed to hatch overnight and arrest at the L1-stage. Synchronized L1s were plated onto 15cm NGM plates seeded with OP50 bacteria. Approximately 8–-10,000 L1s were plated to 15–20 plates for each strain. As N2 cells are only needed to control for the FACS sort, only 5 plates were prepared. These plates were incubated at 20˚C for 72hr to reach adulthood prior to collection.

The entire process of cell preparation and sort was completed in a single day. Animals were washed off plates and spun down and then washed in M9 media to remove bacteria. Typically, pellets for each strain would be ~500μl in volume, and these would be split into two tubes. 750μl of lysis buffer (200μM DTT, 0.25% SDS, 20mM HEPES pH8.0, 3% sucrose) was added to each tube and samples were incubated for approximately 6–7 minutes. Worms were then rapidly washed in M9 five times. Next, 500μl of 20mg/ml freshly made pronase solution was added. Samples were incubated in pronase approximately 20 minutes. Every 2–3 minutes, each sample was disrupted by pipetting with a P200, and samples were monitored for dissociation with a dissection microscope. When large worm chunks were no longer visible under the dissection microscope, the reaction was stopped by adding 250μl ice cold PBS-FBS solution (1X PBS solution with 2% Fetal Bovine Serum). Samples were then centrifuged in the cold at top speed in a microcentrifuge for 10 minutes, and pelleted cells were resuspended in 500μl FBS. Cells were then syringe-filtered (5μm pore). Samples were spun again in the cold and

resuspended in ½ the starting volume of FBS. 80,000–100,000 GFP+ cells were collected for each sample at the UCSD Flow Cytometric Core in Moore's Cancer Center. Cells were sorted directly into Trizol LS and stored at -80˚C until preparation. RNA was isolated using Qiagen RNAeasy kit.

### RNA-seq analyses

RNA library preparation and sequencing were performed at the Institute for Genomic Medicine at UCSD. RNA libraries were prepared with Illumina TruSeq kit. Libraries were sequenced on an Illumina HiSeq4000. Data analyses were performed using the Galaxy platform [27]. We obtained 50–70 million single-end reads/sample. Reads for each sample were mapped and aligned using TopHat [45]. Identification of expressed genes in wild type was determined using Cufflinks. "Expressed" genes were selected by filtering for genes with an FPKM >10 in both replicates. This analysis produced 1,812 transcripts (S2 Table). Using the BioVENN site, our list of "expressed genes" from wild type neurons was compared with those identified as enriched in adult epidermis and muscle and expressed in neurons by RNA-seq, as well as larval A-type motoneurons by microarray [22,23,46]. For differential expression analyses between wild type and *acr-2(gf)* neurons, BAM files were loaded into the HTSEQ program and then analyzed by DESeq2 for differential expression analyses (S2 Table) [26,47]. GO-term enrichment analysis was performed using the GOrilla algorithm for enrichment of Biological Process terms[28]. Data analyses were performed in Microsoft Excel, R, and Graphpad Prism. Sequencing datasets have been deposited in the Gene Expression Omnibus (Accession GSE139212).

### Convulsion and pharmacological analyses

All behavior observations reported here were made on mutations that were outcrossed with N2 for at least 4 times. Convulsions were defined as simultaneous contraction of the body wall muscles producing a rapid, concerted shortening in body length. The convulsion frequency for day-1 adult animals was calculated during a 90-second period of observation. For levamisole sensitivity, ten day-1 adult animals were transferred to fresh plates containing 1 mM levamisole. Animals were scored for paralysis ever 15 minutes for 1 hour. Sensitivity to 500 μM or 1mM aldicarb was assessed by transferring ten day-1 adults to fresh aldicarb plates and by monitoring worms for paralysis every 30 minutes by gently touching the animal with a platinum wire. Drug sensitivity was quantified from at least three independent experiments.

### Imaging and microscopy

Images of fluorescent reporter lines were taken on a Zeiss LSM 700 or 800 confocal microscope. Animals were mounted in thick agarose (10%) and rolled with the ventral side up for consistent imaging. Hyperstacks were processed in ImageJ. (For images in Fig 4, *Pgcy-33::gfp* strains, gains were set at 500V for mKate2 and 500V for GFP. For images in Fig 3, of *juIs14* strains, gains were set at 550V for mKate2 and 600V for GFP.). All images are taken using the 63X objective. Scoring of *Pins-29::gfp* expression in Fig 4 was performed using a Zeiss Axioplan 2 compound microscope at the 63X objective unless otherwise indicated. A neuron was scored as "expressed" if GFP signal was clearly visible in the cell body and dendrites through the eyepiece (with expression in *acr-2(gf)* as a baseline comparison). Images of INS-29::mKate2 in coelomocytes was were captured using a Zeiss AxioImager compound microscope with the 100X objective.

## Molecular biology and *C. elegans* transformation

Transcriptional gfp reporters (pCZ1002, pCZ1003) were made using Gibson assembly into pPD95.75. Vector was cut using restriction enzymes and PCR-amplified promoters were inserted. Approximately 2kb upstream of each neuropeptide gene was used as putative promoter sequence. Primers used for *Pins-29* were (gene-specific sequence in uppercase): Forward 5'tgcatgcctgcaggtcgactCTTTAAAATGGTTAATTTTGTAGTTAG3'/Reverse 5'tggccaatcccggggatcctTTTTTTTATTTCACAATATAATATACTTTATAC3'. Primers used for *Pnlp-1* were: Forward 5'tgcatgcctgcaggtcgactTTGTTTTATCCAACAT TATTCAC3'/Reverse 5'tggccaatcccggggatcctCGTTGCCTCAAGTTGATG3'. To generate mKate2 reporters for neuropeptide expression constructs (pCZ1004, pCZ1005), GFP sequence in pPD95.75 was replaced with SL2-mKate2 sequence. Genomic sequences for *flp-12* or *ins-29* was then amplified from genomic DNA and inserted into the modified pPD95.75 vector using Gibson Assembly. Primers for *ins-29*: Forward 5'aagcttgcatgcctg-caggtCTTTAAAATGGTTAATTTTGTAGTTAG3'/Reverse 5'tgaaagtaggatgaga-cagcTCAAGCAAGATTTGAAGG3'. Primers for *flp-12* were: Forward 5'tgcatgcctgcaggtACAACAAAAGTATTTTTGACG3'/ Reverse 5'agtaggatgaga-cagcCTACTTTCGTCCAAATCG3'. These sequences include the entire coding sequence plus 2kb upstream promoter. To generate the mKate2-tagged *ins-29* construct (pCZ1006), the SL2 sequence was deleted from the *Pins-29::ins-29::SL2::mKate2* construct using the Q5 site-directed mutagenesis kit (New England Biolabs). To generate the *Pgcy-33::acr-2(gf)::gfp* trans-gene (pCZ1007), the *unc-129* promoter from pCZGY1262 generated by Qi *et al.* [37] was replaced with the published *gcy-33* promoter [48] by restriction enzyme digest and ligation. *Pgcy-33* was amplified from genomic DNA using mutagenic primers inserting 5' and 3' restriction enzyme sites HindIII and Pst1, respectively, and used to replace in the *Punc-129* promoter. A list of plasmids used in this study can be found in S1 Table.

cDNAs were generated using the SuperScript RT kit from Invitrogen. 2μg input RNA (from synchronized young adult *acr-2(gf)* populations) was used for each RT reaction. RNA was extracted and isolated using Trizol reagent and chloroform extraction. 1μl of RT (~100ng) was used in each PCR reaction. Nested reactions were used for amplifying both *ins-29* and *ins-25* cDNAs. For *ins-29*, the first reaction used either SL1 or SL2 forward primer with a gene specific reverse primer: 5' gcaagatttgaaggacagcac 3'. In the second reaction 2μl of the first PCR was used with Primers: Forward 5'TTCTGTAAATTTGTATTCCTGATC, Reverse 5' GATTTGAAGGACAGCACAAT 3'. For *ins-25*, the first reaction used either SL1 or SL2 forward primers with a gene specific reverse primer: 5' caaatttgggcaacacatattc 3'. In the second reaction, 2μl of the first PCR reaction was used with Primers: Forward 5' ATGTTGTTCAAAATCATCATT 3', Reverse 5' GGGCAACACATATTCTTCAG 3'. Product using the SL2 primer was only detected for *ins-25* transcript and verified by Sanger sequencing.

*C. elegans* transgenic multi-copy arrays were generated using standard protocols [See S1 Table for list of transgenic strains made in this study] [49]. Reporter DNA was typically injected at 25ng/μl concentration. The *flp-12* over-expression plasmid was injected at 5ng/μl, as injection at 25ng/μl failed to yield transgenics.

## Supporting information

**S1 Fig. INS-29 is secreted in wild type and *acr-2(gf)* animals.** A-F. Shown is expression of directly tagged INS-29 (*Pins-29::ins-29::mKate2, juEx8066*) in a posterior coelomocyte of D1 adults. (A-C) Shown are fluorescent, differential interference contrast (DIC) and overlay, respectively, of a posterior coelomocyte in wild type carrying the *juEx8066* transgene (D-F)

Shown are fluorescent, DIC, and overlay images of a posterior coelomocyte in *acr-2(gf)*. Arrows in all images indicate the larger vacuoles in this cell which contain INS-29::mKate2. Scale bar = 10μm.
(PDF)

**S1 Table. Strains and plasmids used in this study.**
(DOCX)

**S2 Table. Transcriptome analyses of wild type and *acr-2(gf)* neurons.**
(XLSX)

**S3 Table. Aldicarb timecourse for neuropeptide mutants with *acr-2(gf)*.**
(DOCX)

**S4 Table. Aldicarb timecourse for neuropeptide mutants.**
(DOCX)

**S5 Table. Levamisole timecourse for neuropeptide mutants with *acr-2(gf)*.**
(DOCX)

**S6 Table. Levamisole timecourse for neuropeptide mutants.**
(DOCX)

**S7 Table. Levamisole timecourse for *ins-29* over-expression.**
(DOCX)

**S8 Table. Genotyping primers.**
(DOCX)

## Acknowledgments

We thank our lab members for helpful discussions, and Matt Andrusiak and Ngang Heok Tang for comments on the manuscript. We also thank Rachel Kaletsky and Coleen Murphy for sharing advice on neuron isolation, Martin Hudson for the XA2260 strain, and Niels Ringstad for discussion. Some strains were provided by the Caenorhabditis Genetics Center, which is funded by NIH Office of Research Infrastructure Programs (P40 OD010440) and the National Bioresource Project of Japan.

## Author Contributions

**Conceptualization:** Katherine A. McCulloch, Yishi Jin.

**Data curation:** Katherine A. McCulloch.

**Formal analysis:** Katherine A. McCulloch, Kingston Zhou.

**Funding acquisition:** Katherine A. McCulloch, Yishi Jin.

**Investigation:** Katherine A. McCulloch, Kingston Zhou.

**Methodology:** Katherine A. McCulloch, Kingston Zhou.

**Project administration:** Yishi Jin.

**Resources:** Yishi Jin.

**Supervision:** Katherine A. McCulloch, Yishi Jin.

**Validation:** Katherine A. McCulloch.

**Visualization:** Katherine A. McCulloch, Kingston Zhou.

**Writing – original draft:** Katherine A. McCulloch.

**Writing – review & editing:** Katherine A. McCulloch, Yishi Jin.

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
