## [Decision Letter · Decision Letter 0]

6 Jan 2020

PONE-D-19-30924

Neuronal transcriptome analyses reveal novel neuropeptide modulators of excitation and inhibition imbalance in C. elegans

PLOS ONE

Dear Dr. Jin,

Thank you for submitting your manuscript to PLOS ONE. After careful consideration, we feel that it has merit but does not fully meet PLOS ONE’s publication criteria as it currently stands. Therefore, we invite you to submit a revised version of the manuscript that addresses the points raised during the review process.

As you can see, while both reviewers appreciate the interest of the work, they have serious concerns about the absence of key supportive experiments and, arising from this, the interpretations presented. Essential data requested include details of the expression of manipulated genes (e.g. ins-29) and mutants/knockdown in specific cell types (e.g. BAG neurons). Additional substantive mechanistic evidence is essential given the highly unusual scenario proposed whereby increased activity of Acr-2 induces enhanced expression of the acr-2 locus.

We would appreciate receiving your revised manuscript by Feb 20 2020 11:59PM. To enhance the reproducibility of your results, we recommend that if applicable you deposit your laboratory protocols in protocols.io, where a protocol can be assigned its own identifier (DOI) such that it can be cited independently in the future. For instructions see: http://journals.plos.org/plosone/s/submission-guidelines#loc-laboratory-protocols

We look forward to receiving your revised manuscript.

Kind regards,

Brian D. McCabe, Ph.D.

Academic Editor

PLOS ONE

Journal Requirements:

Reviewers' comments:

Reviewer's Responses to Questions

**Comments to the Author**

1. Is the manuscript technically sound, and do the data support the conclusions?

Reviewer #1: Partly

Reviewer #2: Yes

2. Has the statistical analysis been performed appropriately and rigorously? 

Reviewer #1: I Don't Know

Reviewer #2: Yes

3. Have the authors made all data underlying the findings in their manuscript fully available?

Reviewer #1: Yes

Reviewer #2: No

4. Is the manuscript presented in an intelligible fashion and written in standard English?

Reviewer #1: Yes

Reviewer #2: Yes

5. Review Comments to the Author

Reviewer #1: In this paper, the authors asked whether upregulation of cholinergic activity would induce homeostatic responses in the cholinergic neurons, counteracting the increased activity. To this end, they used a gain-of-function mutation in a neuronal acetylcholine receptor in C. elegans, acr-2(gf), which induces convulsions in the animal. The authors assessed which genes are upregulated, by RNAseq. They found that several neuropeptide genes were upregulated, and studied the expression pattern and the functional effects of the most upregulated peptides in modulating / counteracting function of acr-2(gf). ins-25, ins-29, flp-12 and nlp-1 were the most upregulated neuropeptide genes. INS-29 is expressed, unexpectedly, in a gas sensory neuron, BAG. The authors then analyzed whether loss of function of these peptides would lead to a change in the convulsions induced by acr-2(gf). They find that single mutations of the candidate neuropeptide genes did not have any effects, but triple-deletion of ins-29, ins-25 and flp-12 reduced the convulsions. This is surprising, as if anything, one would have expected that the overexpressed peptides are inhibitory, and that in their absence, there should be more convulsions. Actually, when the authors overexpressed flp-12 and ins-29, this reduced the convulsions.

The work presented in the paper is technically sound and the questions is of interest. The findings are interesting and I do not doubt the work per se, however, the current state does not allow a conclusive understanding of what is going on. The logic of the findings is not making sense, as the induced hyperactivity of the cholinergic system would be expected to induce upregulation of systems that can counterbalance this hyperactivity, i.e. classical homeostatic responses to allow maintaining somewhat normal functionality of the nervous system. However, knock out of the upregulated neuropeptide genes caused a reduction of the convulsions rather than further increase. The authors should comment on this paradox in the discussion, particularly as overexpression also caused reduction of convulsion (which is more in line with expectations).

Altering the function of neuropeptide signaling often results in surprising findings that are not easily reconciled. At stage, the work is premature and asks for more experiments to understand the precise nature of the phenotypes. Possibly, the imbalance in the motor nervous system is more complex due to the effects on GABAergic signaling, which is usually co-stimulated when cholinergic neurons are active. Also, have the authors considered that genes downregulated may have an important role here, e.g. because they are normally promoting (cholinergic) nervous system function?

Specific points:

The authors say that they analyzed the expressome of adult cholinergic neurons. This is imprecise, because they used the acr-2 promoter-driven GFP to isolate cells for RNAseq. However, while acr-2 is expressed in many cholinergic motor neurons, it is also expressed in numerous unidentified cells, which are not necessarily cholinergic. For example, BAG, in which the INS-29 neuropeptide is overexpressed (as well as ACR-2), is a glutamatergic neuron (Pereira et al., 2015, eLife). Thus, the wording should be adjusted.

The authors show that acr-2 is expressed in BAG neurons, and that acr-2 expression itself is upregulated in BAG in the acr-2(gf) background. This overexpression is surprising – why would the intrinsic acr-2 locus respond to hyperactivity of its own product by upregulation? One would rather expect a downregulation. In this context: Does ACR-2 activity in BAG lead to increased INS-29 release due to depolarization of BAG? Upregulation of a peptide generally counteracting cholinergic hyperactivity, but in a neuron outside the cholinergic system, implies that some signal from these motor neurons is released in response to their hyperactivity, which is sensed in BAG and leads to upregulation of INS-29. This would then lead to increased release of INS-29 peptides. However, if ACR-2 itself is expressed in BAG neurons, increased INS-29 release may not be in response to acr-2(gf) in the cholinergic system but simply because acr-2 is more active in BAG and the cell is more depolarized. To test this the authors could knock down acr-2(gf) in BAG neurons, cell-specifically, and test if these animals still show compensatory activity of ins-29.

In Fig 5C, most animals of ins-29 ins-25(0); flp-12(0); acr-2(gf) were paralyzed. However, they were almost non-paralyzed in Table S3 (60 min time point).

Minor points:

1) In Fig. 4B, there is expression of gcy-33 in the dorsal cord. This is surprising as gcy-33 expression was previously described only in BAG, URX, AQR and PQR – do the authors know which cells these are?

2) In Fig 5A, flp-24(0); nlp-12(0) and ins-29 ins-25(0); flp-12(0) partially blocked acr-2(gf). Did they affect cholinergic function also in a normal nervous system (i.e., without acr-2(gf)?

3) In Fig 5D, can the phenotype of ins-29 ins-25(0); flp-12(0); acr-2(gf) be rescued by flp-12 or ins-29 overexpression?

Reviewer #2: In this aper, the authors describe their transcriptomic analysis of the BAG neurons and up-regulation of one of the insulin-like genes ins-29 in the acr-2(gf) background. This finding seams interesting.

However, the authors did not examine how the ins-29 up-regulation is relevant to the phenotype of acr-2(gf).a The authors bad better show the following points.

1. Where (in which neurons) is the ins-29 originally expressed?

The transgene ins-29p::gfp (or other reporters) can address this question.

2. Is the up-regulation of ins-29 expression responsible for the phenotype of acr-2(gf)?

The gene disruption of ins-29 (using the tm1922 mutant) in acr-2(gf) can easily address this question.

In addition, in the double mutant above, a rescue experiment of ins-29 in a BAG neurons-specific manner

can enforce the responsibility of ins-29.

3. Is the ins-29 translated and secreted after over transcription in BAG neurons under the acr-2(gf) background?

The transgene acr-2p::ins-29::mcherry can address this question.

Otherwise, the authors should explain how the over transcription of ins-29 itself induce the phenotype of the

acr-2(gf) mutant.

After the address of authors to my comments, I would like to accept their paper.

6. PLOS authors have the option to publish the peer review history of their article (what does this mean?). If published, this will include your full peer review and any attached files.

Reviewer #1: No

Reviewer #2: No

---

## [Author Response · Author response to Decision Letter 0]

30 Mar 2020

please see the file 'response to reviewers'

---

## [Decision Letter · Decision Letter 1]

18 May 2020

Neuronal transcriptome analyses reveal novel neuropeptide modulators of excitation and inhibition imbalance in C. elegans

PONE-D-19-30924R1

Dear Dr. Jin,

We are pleased to inform you that your manuscript has been judged scientifically suitable for publication and will be formally accepted for publication once it complies with all outstanding technical requirements.

With kind regards,

Brian D. McCabe, Ph.D.

Academic Editor

PLOS ONE

Additional Editor Comments (optional):

Reviewers' comments:

Reviewer's Responses to Questions

**Comments to the Author**

1. If the authors have adequately addressed your comments raised in a previous round of review and you feel that this manuscript is now acceptable for publication, you may indicate that here to bypass the “Comments to the Author” section, enter your conflict of interest statement in the “Confidential to Editor” section, and submit your "Accept" recommendation.

Reviewer #1: All comments have been addressed

2. Is the manuscript technically sound, and do the data support the conclusions?

Reviewer #1: Partly

3. Has the statistical analysis been performed appropriately and rigorously? 

Reviewer #1: Yes

4. Have the authors made all data underlying the findings in their manuscript fully available?

Reviewer #1: No

5. Is the manuscript presented in an intelligible fashion and written in standard English?

Reviewer #1: No

6. Review Comments to the Author

Reviewer #1: all fine, thank you for addressing my comments, the authors made a good job and the paper may now be accepted

7. PLOS authors have the option to publish the peer review history of their article (what does this mean?). If published, this will include your full peer review and any attached files.

Reviewer #1: No

---

## [Editor Report · Acceptance letter]

22 May 2020

PONE-D-19-30924R1 

Neuronal transcriptome analyses reveal novel neuropeptide modulators of excitation and inhibition imbalance in *C. elegans*

Dear Dr. Jin:

I am pleased to inform you that your manuscript has been deemed suitable for publication in PLOS ONE. Congratulations! Your manuscript is now with our production department. 

With kind regards,

on behalf of

Dr. Brian D. McCabe 

Academic Editor

PLOS ONE